# The Effects of Wearing Facemasks during Vigorous Exercise in the Aspect of Cardiopulmonary Response, In-Mask Environment, and Subject Discomfort

**DOI:** 10.3390/ijerph192114106

**Published:** 2022-10-28

**Authors:** Juntaek Hong, Juahn Byun, Joong-on Choi, Dain Shim, Dong-wook Rha

**Affiliations:** Department and Research Institute of Rehabilitation Medicine, Yonsei University College of Medicine, Seoul 03722, Korea

**Keywords:** facemask, high intensity exercise, cardiopulmonary, in-mask environment

## Abstract

Non-pharmaceutical intervention such as wearing a mask during the pandemic of SARS-CoV-2 is one of the most important ways to prevent the spread of the virus. However, despite high effectiveness and easy to access, the biggest problem is ‘discomfort’. The purpose of this study was to measure the changes of cardiopulmonary response and related factors affecting breathing discomfort when wearing a mask during vigorous exercise. Fifteen healthy male adults participated in this study. The experimental protocol consisted of three conditions: no mask; KF-94 mask; and sports mask. Each condition consisted of three stages: stage I, 2 m/s on even level; stage II, 2 m/s with 5° inclination; and stage III, 3 m/s on even level. Oxygen saturation (SaO_2_) and heart rate (HR), partial pressure of carbon dioxide (pCO_2_), energy expenditure index (EEI), in-mask temperature, humidity, and a five-point scale questionnaire to evaluate subjective discomfort were measured. The results show that there was a significantly higher discomfort score in mask conditions compared with no mask (*p* < 0.05) and only pCO_2_ change significantly related to subjective discomfort during exercise (*p* < 0.05). Moreover, the pCO_2_ washout was significantly disturbed when wearing a sports mask in stages 2 and 3, which was related to wearer subjective discomfort

## 1. Introduction

Over the past two years, coronavirus disease 2019 (COVID-19) has caused more than 530 million cumulative infections and over 6 million deaths worldwide. Currently, the impact of COVID-19 is still continuing, with more than 500,000 new confirmed cases per day as of June 2022 [1]. Moreover, the risk of infectious disease outbreaks could be increasing due to various global changes such as technological, demographic, and climatic change despite improvement in the overall level of health care [2].

Meanwhile, wearing a mask is one of the most effective ways to lower infection rates by preventing the spread of pathogens and acquisition of the disease [3]. The N95 respirator, a well-known mask, has one of the most effective droplet blocking systems with a filtering efficiency that can block 95% of 0.3 micron-sized particles [4]. Through recent research [5], the KF-94 mask has been shown to have similar efficiency for blocking virus particles as the N95 mask, which has been widely used in the Republic of Korea. In addition to these masks, there are many other types of masks, such as surgical masks, cloth masks, and sports masks. Among them, the sports mask is a type of cloth mask whose main ingredient is polyester, which has the advantage of good wearability and reusability after washing. Although these types of masks have a lower filtering capacity than N95 or KF-94 masks [3,5], this is uncontroversial due to their superior effectiveness compared to not wearing a mask [6]. 

However, despite high effectiveness compared to accessibility, the biggest problem with mask wearing is ‘discomfort’. According to the results of a survey conducted with 4.8 million people [7,8] 44% of responses to the question “Why don’t you wear a mask?” were “because it is uncomfortable”. A number of previous studies have attempted to evaluate the effect of wearing masks from the point of view that the cause of discomfort could be due to changes in the cardiopulmonary response. However, there has been no clear answer. Most previous studies focused on changes in physiologic factors such as oxygen saturation and end-tidal concentration of carbon dioxide, finding no significant change during exercise; however, their protocol designs included mild to moderate intensity exercise, such as comfortable walking speed or brisk walking speed [9,10]. Meanwhile, only a few studies have considered high intensity exercise; Michelle et al. [11] and Salati et al. [12] found N95 mask wearing could increase the end-tidal concentration of carbon dioxide although other parameters such as heart rate and blood pressure [13] and oxygenation index [14] did not show any difference compared to no-mask condition. However, none of these studies under the high intensity exercise measured the partial pressure of carbon dioxide which could reflect energy metabolism in human body more directly.

On the other hand, there were other attempts to analyze the other variables including degree of subjective discomfort as well as the cardiopulmonary response. There are few review articles that have evaluated the comprehensive effect of wearing masks, including subjective discomfort and the work of breathing [15,16,17]. One of the studies referenced in this article proposed that there was a significant relationship between subjective discomfort and in-mask environment, such as humidity and temperature, but the respirator used in this experiment was a respiratory protective mask, not one that would be used in daily life [18]. 

The purpose of this study was to measure the change in cardiopulmonary parameters as well as in-mask environment during high-intensity exercise with various types of masks and evaluate the correlation between subjective breathing discomfort and measured parameters. 

## 2. Materials and Methods

### 2.1. Participants

Participants were healthy and without any history of musculoskeletal or cardiopulmonary disease. Sample size was calculated based on the previous similar study about vigorous exercise performance when wearing a facemask [14]. With the coefficient of variation and expected change, we required 13 participants with an alpha value of 0.05 and power of 90%. All the participants were enrolled through the voluntary recruitment process of clinical research at our college of medicine. Ethical approval was granted by the institutional review board of our institute (2021-1745-001). All participants were informed about the purpose and protocol of the study before enrollment. Written informed consent was obtained from all participants.

### 2.2. Experiment Protocol: Three Trial-Three Stage Protocol

This was an experimental study and the experimental protocol consisted of three trials: no mask; tri-fold shape KF-94 mask composed of nonwoven fabric; and sports mask composed of 93% polyethylene and 7% spandex fabric. All trials were conducted on a treadmill device that enabled the addition of dynamic pitch, gait speed change, or control of both variables simultaneously (M-gait, Motek, Amsterdam, The Netherlands). The three trials were randomly sequenced, and at least 20 min of rest was taken between trials. The next trial was conducted after confirming that the resting heart rate was restored and there were no respiratory symptoms.

Each trial consisted of three stages, with each stage providing a vigorous physiological burden under different gait conditions. In the first stage, at a speed of 2 m/s, the pitch angle was set to 0° and the experiment was performed for 6 min. In the second stage, the speed was the same, but the pitch angle was set to 5° and then performed for 3 min. The last stage was performed for 3 min with pitch angle set to 0° and speed to 3 m/s. Each stage proceeded continuously without a break, and a verbal warning was given 10 s before the speed or the degree of inclination changed to prevent falling. If the participant judged that it was not possible to continue the exercise, it could be stopped immediately, and all data measured at that stage were excluded from the results. The schematic flow of the protocol is shown in Figure 1.

### 2.3. Data Acquisition

Baseline data were collected after resting for at least 15 min while sitting in a chair with a backrest. The experiment was initiated after confirming that there were no specific cardiopulmonary symptoms. Each parameter was measured every minute, and all measured values at each stage were averaged and used as a representative value for each stage. All indicators measured during the experiment are shown in Figure 2.

The cardiopulmonary parameters used in this study were peripheral oxygen saturation (SpO_2_), partial pressure of carbon dioxide (pCO_2_), and heart rate (HR). SpO_2_ and HR were measured using pulse oximetry (uMEC10, Mindray Bio-Medical Electronics Co. Ltd., Shenzhen, China), which was tightly fixed to the fingertip to ensure good contact during exercise. To measure the partial pressure of carbon dioxide, a noninvasive monitoring device (V-Sign Sensor 2; SenTec AG, Arlesheim, Switzerland), which could provide accurate results compared to conventional blood gas analysis [19], was applied to the earlobe according to the manufacturer’s recommendations.

Energy expenditure index (EEI), also called physiologic cost index, is an index that can be easily to measure the degree of energy consumption using only heart rate and walking speed [20]. This index can be used in various gait conditions irrespective of walking speed and gait stability [21,22,23]. The formula used is as follows:$$EEI=\frac{Walking\;Heart\;Rate-Resting\;Heart\;Rate}{Walking\;speed\left(m/min\right)}$$

The temperature and humidity in the mask were measured using a portable temperature and humidity data logger (Hydrochron iButton^®^ DS1923, Maxim Integrated, San Jose, CA, USA). The measurement range was −20 to +85 °C and 0–100%. After measuring room temperature and humidity, the device was attached inside the mask during exercise, and humidity and temperature were measured at 1-min intervals to a resolution of 0.5 °C and 0.6% respectively. To minimize the effect of environment, the experiment was conducted between 4 and 9 p.m., and after confirming that the temperature in the laboratory was between 20 and 27 degrees and the humidity was less than 50%.

Degree of subjective discomfort at each stage was measured by a discomfort questionnaire, which consisted of a five-point scale, based on a previous study [18] that evaluated subject discomfort when wearing a respiratory protective mask. Scores were as follows: (1) comfortable, (2) slightly uncomfortable, (3) uncomfortable, (4) very uncomfortable, and (5) intolerable. All participants were asked to complete the form immediately after the end of each trial.

### 2.4. Statistical Analysis

All statistical analyses were performed using R Studio Statistical Software (version 2022.02.0; R Foundation for Statistical Computing, Vienna, Austria). To compare the differences between the three stages, the Wilcoxon signed-rank test or paired *t*-test was performed depending on the normality of variable distributions. 

A linear mixed model analysis was used to assess the correlation between the measured indices and subjective rating scores from participants for each stage, which considered the within-subject correlation effect due to repeated measurements within participants. Since the independent variables of interest, SpO_2_, pCO_2_, heart rate, EEI, temperature, and humidity in the mask were time-dependent covariates that could change at each measurement point, the intra-individual and inter-individual effects were separated for analyses [24].

## 3. Results

Fifteen healthy male adults participated in this study. General characteristics of the participants in this study were: mean age, 24.30 ± 3.82 years old (21–33 years old); mean height, 175.60 ± 6.65 cm (158–187 cm); mean body weight, 73.53 ± 11.71 kg (53–90 kg). In this study, 135 stages were attempted by conducting three trials with nine stages for 15 participants. However, eight stages (two of stage II and six of stage III) were excluded as the participant requested to stop due to shortness of breath. The number of stages where data could not be measured due to monitoring device failure during exercise were as follows: 13 for SpO_2_, 9 for pCO_2_, 5 for heart rate and EEI, 5 for temperature, and 4 for humidity.

### 3.1. Comparison with Baseline Data

Compared to baseline, SpO_2_ decreased significantly in stages II and III of all trials. pCO_2_ decreased significantly in stages II and III of the trial without a mask, which was considered a washout effect. However, pCO_2_ in trials with KF-94 and sports masks showed no significant changes. In-mask humidity and temperature significantly increased compared with room environment and maintained a relatively steady state in all trials with both masks. Heart rate significantly increased in all stages of the trials, and maximal heart rate was achieved in stage III using the Karvonen formula. Moreover, EEI showed a tendency to increase as the stage went up in all trials and was close to the level seen when performing high-intensity exercise [21]. All these data are shown in Table 1.

### 3.2. Comparison between Groups without Mask, with KF-94 Mask, and with Sports Mask

In the case of SpO_2_, HR, and EEI, there were no significant differences between the three trials with and without masks. However, when wearing a sports mask, the pCO_2_ level was significantly higher in stages II and III than that in the trial without a mask. While in-mask temperature of the sports mask group significantly increased compared to that of the KF-94 mask, in-mask humidity was not significantly different between mask groups (Figure 3).

### 3.3. Subjective Ratings of Discomfort

Subjective ratings of discomfort progressively increased according to the stage of the trials (Table 2). Significantly more discomfort was observed in both trials with KF-94 and sports masks compared to the no mask condition (*p* < 0.05). However, there were no significant differences in discomfort ratings between KF-94 and sports mask trials (Figure 4).

### 3.4. Correlation between Physiologic Parameter and Subjective Discomfort

There was no correlation between the discomfort questionnaire scores and cardiopulmonary parameters. However, in the case of the within effect, the subjective discomfort score was significantly affected by the pCO_2_ change from baseline (*p* < 0.05) (Table 3).

## 4. Discussion

This is the first study to evaluate the effect of mask wearing on cardiopulmonary parameters and the in-mask environment during high-intensity exercise, and the correlation between these parameters and subjective discomfort. SpO_2_ significantly decreased in all trials with or without a mask. This result contrasted with a previous study of moderate intensity exercise with a mask [10]. However, in another study of acute aerobic exercise, it was shown that carbon dioxide production increases, lactic acid builds up, and blood pH increases, which could induce a decrease in hemoglobin oxygen saturation despite the same partial oxygen pressure during exercise, based on the hemoglobin oxygen dissociation curve [25]. Considering the commonly accepted minimum clinically important difference (MCID) for SpO_2_ of ±4 percentage points [26], this result does not represent a significant change.

HR during exercise was significantly higher than that at rest. As the stage increased, HR gradually increased and reached near the maximal HR derived from the Karvonen formula in stage III of all trials, the intended exercise intensity. However, there was no difference between each stage of the three trials with and without masks. This could be because mask wearing did not influence the degree of increase in HR. These results were similar to those of previous studies regarding the effect of wearing masks on HR [27,28]. In another study [29] that measured cardiac output using impedance cardiography, which reflects cardiac response more accurately than HR, there was no difference in this parameter with or without a mask even under vigorous exercise.

In-mask temperature and humidity significantly increased when wearing masks compared with the room environment. Additionally, in-mask temperature of the sports mask group significantly increased compared to that of the KF-94 mask. Recent studies have suggested that thermal and evaporative resistances of the face masks could be more affected by variables such as the fabric system, mask design, size and fitting, and the volume of the airgap of the mask itself other than filtering efficacy [30,31]. Future studies would be needed while controlling for these variables. Trials wearing a mask showed a significantly increased degree of discomfort compared with the trials without a mask. These results agree with those of previous studies [32,33]. However, our study showed no statistically significant relationship between subjective discomfort and focal changes in the in-mask environment assessed by a linear mixed model, in contrast to a previous study [18] that concluded that subjective breathing discomfort and in-mask environmental change were significantly correlated. The previous study was conducted by artificially changing the temperature and humidity in the mask using a controller connected to the mask. However, in this study, exercise-induced in-mask temperature and humidity increased and reached a steady state very early, even before the subjects felt discomfort. Moreover, the previous study did not measure other variables that could affect subject discomfort such as pCO_2_ and SaO_2_. The submaximal intensity of exercise could also have influenced the difference between studies.

While pCO_2_ decreased as the stages increased when the mask was not worn, there was no significant difference in pCO_2_ among the stages when wearing a mask in this study. This means that wearing masks, especially sports masks, could disturb CO_2_ washout. This finding was contrary to previous reports that CO_2_ concentration in the bloodstream was not elevated when wearing masks [11,34]. However, it has also been reported that in-mask CO_2_ concentration increased during exercise and mask breathing caused excessive CO_2_ inhalation approximately seven times greater per breath than normal breathing in an N95 respirator [12]. The increase in CO_2_ concentration in the mask may have inhibited CO_2_ washout during ventilation in this study. However, there have been few comparative studies of carbon dioxide concentration according to the mask types, only comparisons under mild intensity [35]. Further study will be needed to clarify the causes that affect CO_2_ concentration between types of masks.

In addition, the change in pCO_2_ was most highly correlated with subjective discomfort score in this study. There are some explanations for why CO_2_ washout increases during high-intensity exercise. One of the most common explanations is the effect of respiratory compensation, which induces respiratory alkalosis to prevent metabolic lactic acidosis caused by the failure of the body’s buffering mechanism [36]. This could explain the mechanism of maintaining blood CO_2_ concentrations, but it is insufficient to explain the effect of generating hypocapnia by hyperventilation. According to a previous study [37] the actual breathing rate was much higher than the ventilation demand to maintain the buffer system under vigorous exercise. Moreover, voluntary hypocapnic hyperventilation could affect not only acid-base balance but also the body temperature regulation system by facilitating a cutaneous vasodilatory response [38]. Considering this phenomenon, the failure of voluntary hypocapnia when wearing a mask might induce a body temperature regulation problem and consequently more breathing effort, which could lead to breathing discomfort. However, sample size in this study was relatively small, so further study would be needed to prove this result.

This study has some limitations. First, we recruited only healthy young adults. Further studies are needed with older people who could be more vulnerable to ventilation when wearing masks. Second, the subjective discomfort questionnaire is too simple. In future research, more detailed questionnaires about the severity and type of discomfort would be helpful in clarifying the characteristics of discomfort induced by wearing masks. Third, respiratory rate, which may reflect the hypocapnic hyperventilation, and end tidal CO_2_ which related to partial pressure of CO_2_, was not measured in this study. Fourth, we did not control the other variables which could affect in-mask environment such as mask style, fitting, or the volume of airgap in the mask.

## 5. Conclusions

In conclusion, when comparing trials of vigorous exercise with and without a mask, in-mask humidity and temperature increased significantly, but there were no differences in heart rate, SpO_2_, and EEI. In contrast, CO_2_ washout was significantly disturbed when wearing masks, especially sports masks, which was related to wearer subjective discomfort.

## Figures and Tables

**Figure 1 ijerph-19-14106-f001:**
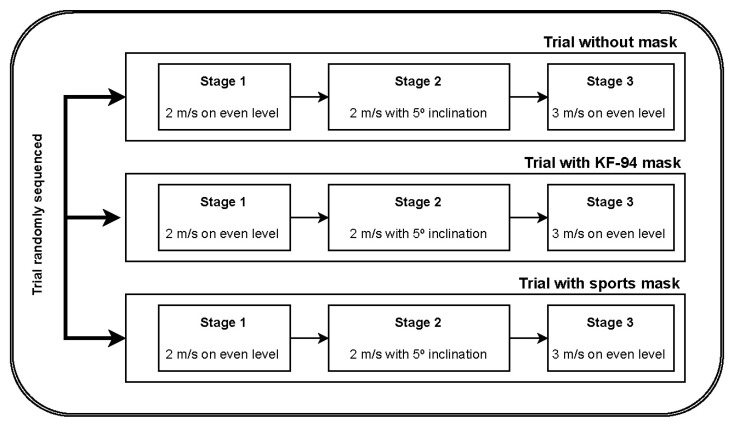
Schematic flow of experiment protocol; 3 trials with 3-stage protocol.

**Figure 2 ijerph-19-14106-f002:**
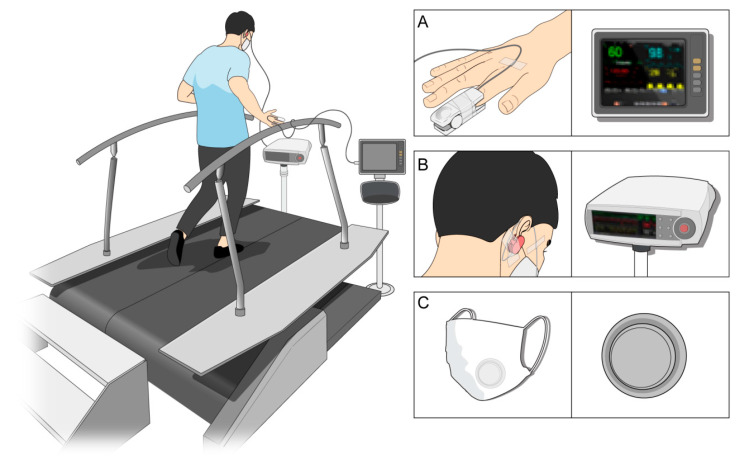
Schematic drawing of the data acquisition system. (**A**) Infrared transcutaneous sensor attached to the fingertip to measure peripheral oxygen saturation and heart rate; (**B**) transcutaneous monitoring sensor applied to the earlobe to measure partial pressure of carbon dioxide; (**C**) portable temperature and humidity data logger attached inside the mask.

**Figure 3 ijerph-19-14106-f003:**
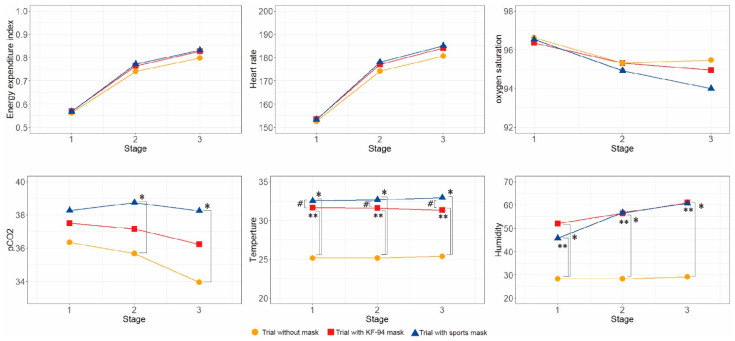
Comparison of physiologic parameters among KF-94 mask, sports mask, and no mask conditions. Data were analyzed using the paired *t*-test or Wilcoxon signed-rank test depending on the normality compared between conditions. * *p* < 0.05 when comparing trials between no mask and sports mask conditions, ** *p* < 0.05 when comparing trials between no mask and KF-94 mask conditions, # *p* < 0.05 when comparing trials between KF-94 mask and sports mask conditions.

**Figure 4 ijerph-19-14106-f004:**
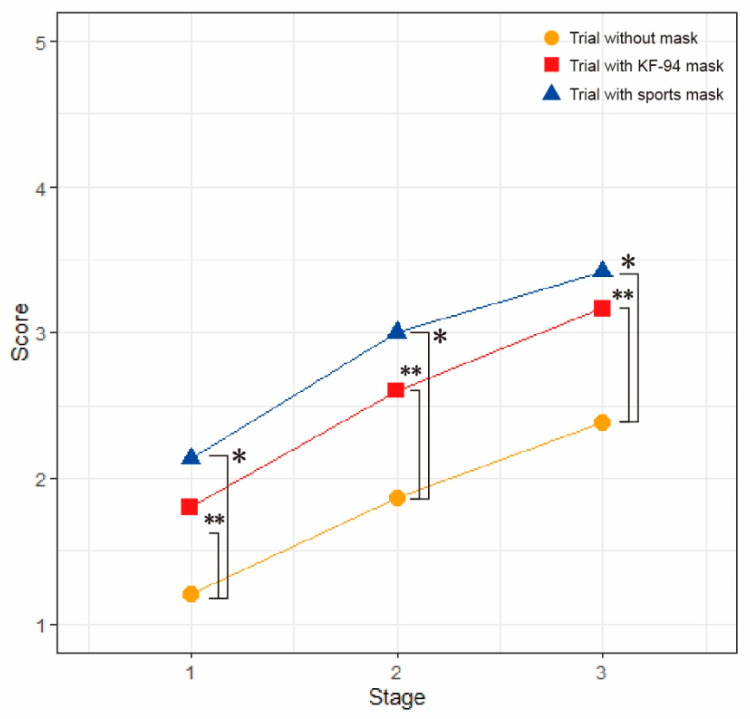
Comparison of subjective ratings of discomfort among KF-94 mask, sports mask, and no mask conditions. Data were analyzed using the paired *t*-test or Wilcoxon signed-rank test depending on the normality compared between conditions. * *p* < 0.05 when comparing trials between no mask and sports mask conditions, ** *p* < 0.05 when comparing trials between no mask and KF-94 mask conditions.

**Table 1 ijerph-19-14106-t001:** Physiologic parameters in three conditions of mask wearing compared to baseline.

		SaO_2_ ^1^		pCO_2_ ^2^		Heart Rate		Temperature		Humidity		EEI ^3^
		Mean ± SD	*p*-Value	Mean ± SD	*p* Value	Mean ± SD	*p*-Value	Mean ± SD	*p*-Value	Mean ± SD	*p*-Value	Mean ± SD
Baseline		96.93 ± 1.03		38.61 ± 4.14		88.20 ± 12.72		25.14 ± 0.90		28.27 ± 9.88		
No mask	Stage 1	96.61 ± 1.69	0.640	36.35 ± 3.63	0.066	152.57 ± 11.41	<0.001 ***					0.56 ± 0.09
	Stage 2	95.31. ± 2.53	0.016 **	35.67 ± 4.45	0.022 *	175.03 ± 9.90	<0.001 ***					0.75 ± 0.10
	Stage 3	95.46 ± 1.47	0.028 *	33.94 ± 5.15 *	0.003 *	180.65 ± 12.51	<0.001 ***					0.80 ± 0.11
KF- 94 mask	Stage 1	96.34 ± 1.37	0.129	37.50 ± 2.55	0.679	153.45 ± 10.95	<0.001 ***	31.66 ± 0.92	<0.001 ***	51.97 ± 13.35	<0.001 ***	0.57 ± 0.09
	Stage 2	95.31. ± 1.48	0.002 *	37.15 ± 3.57	0.547	177.37 ± 10.56	<0.001 ***	31.57 ± 0.77	<0.001 ***	56.38 ± 11.77	<0.001 ***	0.77 ± 0.11
	Stage 3	94.95 ± 1.61	0.001 **	36.23 ± 4.29	0.082	184.05 ± 7.39	<0.001 ***	31.34 ± 0.72	<0.001 ***	60.89 ± 11.80	<0.001 ***	0.83 ± 0.09
Sports mask	Stage 1	96.49 ± 1.26	0.169	38.26 ± 3.96	0.810	153.31 ± 10.81	<0.001 ***	32.53 ± 1.34	<0.001 ***	45.62 ± 8.94	<0.001 ***	0.57 ± 0.10
	Stage 2	95.11. ± 1.88	0.015 *	38.73 ± 4.82	0.974	178.13 ± 10.59	<0.001 ***	32.66 ± 0.69	<0.001 ***	56.72 ± 12.10	<0.001 ***	0.77 ± 0.11
	Stage 3	94.00 ± 2.85	0.004 **	38.25 ± 6.04	0.684	185.05 ± 8.05	<0.001 ***	32.75 ± 1.76	<0.001 ***	60.68 ± 11.41	<0.001 ***	0.83 ± 0.11

Data were analyzed using the paired *t*-test or Wilcoxon signed-rank test depending on the normality compared to the baseline parameter, * *p* < 0.05, ** *p*< 0.01, *** *p*< 0.001, ^1^ oxygen saturation, ^2^ partial pressure of carbon dioxide, ^3^ energy expenditure index.

**Table 2 ijerph-19-14106-t002:** Subjective ratings of discomfort during exercise in three conditions of wearing masks.

	Stage 1	Stage II	Stage III
	Mean ± SD	Mean ± SD	Mean ± SD
Trial no mask	1.20 ± 0.41	1.87 ± 0.74	2.38 ± 1.04
Trial with KF-94 mask	1.80 ± 0.41	2.60 ± 0.74	3.17 ± 1.03
Trial with sports mask	2.13 ± 0.74	3.00 ± 0.65	3.42 ± 1.00

**Table 3 ijerph-19-14106-t003:** Correlations between physiologic parameters and subjective discomfort ratings.

	Subject Rating of Degree of Discomfort during Stage			
	Between Effect		Within Effect	
	Beta Coefficient (SE)	*p*-Value	Beta Coefficient (SE)	*p*-Value
SaO_2_ ^1^	−0.018(0.116)	0.8812	−0.0015(0.039)	0.6989
pCO_2_ ^2^	0.046(0.054)	0.4098	0.045(0.019)	0.0167 *
Heart rate	0.003(0.011)	0.814	−0.002(0.009)	0.7813
EEI ^3^	0.014(0.011)	0.1803	−0.007(0.009)	0.4315
Temperature in mask	0.104(0.151)	0.5036	−0.07(0.055)	0.2058
Humidity in mask	−0.013(0.029)	0.657	−0.002(0.007)	0.8218

Data were analyzed using linear mixed model analyses. SE, standard error; * *p* < 0.05. ^1^ oxygen saturation, ^2^ partial pressure of carbon dioxide, ^3^ energy expenditure index.

## Data Availability

The data presented in this study are available upon request from the corresponding authors. The data are not publicly available for reasons related to participant privacy.

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
