# Peer review of "The Effects of Wearing Facemasks during Vigorous Exercise in the Aspect of Cardiopulmonary Response, In-Mask Environment, and Subject Discomfort"

_ijerph, 2022, doi:10.3390/ijerph192114106_

Round 1

Reviewer 1 Report

Thank you for giving me the opportunity to review this paper. Generally, the experiment was straightforward, and limited data were presented in this paper. Only 6 variables were presented with 3 conditions (without mask/ sports mask/ KF94) were tested.

Section 2 -Materials and methods need to be written in more details.

1.      Selection of the participants should be described in detail. The recruitment process should be described in more detail. For example, was it based on advertisement/ voluntarily/ or randomly selected?

2.      The minimum sample size for this study should be indicated based on the sample size calculation

3.      Section 2.1 – mean and SD should be described in results section.

4.      Section 2.2 – This was an experiment study done in the laboratory.

E   Environmental characteristics in the lab should be presented (i.e. environment temperature, humidity, Data collection time (AM or PM), etc.)  

Line 77 page 2 – “The three trials were randomly sequenced, with sufficient rest allowed between each trial for heart rate to return to baseline”- Please indicate -on average how long time taken for rest and for the heart rate to return to baseline?

Section 2.3 Please indicate about the ethical clearance and whether informed consent was collected from each participant in this study.

Discussion

Please explain and discuss why the results of this study indicated CO2 washout was significantly disturbed when wearing sports masks compared to KF94? In the introduction the authors indicated that sports masks are designed for convenience of wearing and have a lower filtering capacity than N95 or KF-94 masks. But the results of this study indicated “in-mask temperature of the sports mask group significantly increased compared to that of the KF-94 mask”.

In line with the above issue, the detail information about materials and condition of both masks (sports mask vs KF94) should also be presented in more details in methodology section.

Please improve and describe in more detail about the limitations of this study.

Page 8 line 273 – Please confirm about the inform consent statement written in this paper.   

References – a lot of papers listed were not current (latest 5 years).

Reviewer 2 Report

The main problem I see with the study is that there is a low number of participants that could have been much larger. I see another lack of the study in the fact that a sufficient number of studies in this area. With better study design have already been published

Driver S, Reynolds M, Brown K, et al. Effects of wearing a cloth face mask on performance, physiological and perceptual responses during a graded treadmill running exercise test Br J Sports Med 2022;56:107–113.

Shaw, K. A., Zello, G. A., Butcher, S. J., Ko, J. B., Bertrand, L., & Chilibeck, P. D. (2021). The impact of face masks on performance and physiological outcomes during exercise: a systematic review and meta-analysis. Applied physiology, nutrition, and metabolism = Physiologie appliquee, nutrition et

Egger, F., Blumenauer, D., Fischer, P., Venhorst, A., Kulenthiran, S., Bewarder, Y., Zimmer, A., Böhm, M., Meyer, T., & Mahfoud, F. (2022). Effects of face masks on performance and cardiorespiratory response in well-trained athletes. Clinical research in cardiology : official journal of the German Cardiac Society, 111(3), 264–271. https://doi.org/10.1007/s00392-021-01877-0

Asín-Izquierdo, I., Ruiz-Ranz, E., & Arévalo-Baeza, M. (2022). The Physiological Effects of Face Masks During Exercise Worn Due to COVID-19: A Systematic Review. Sports health, 14(5), 648–655. https://doi.org/10.1177/19417381221084661

Is not suitable sentence for the abstract „However, despite high effectiveness, the biggest problem with mask wearing is low compliance.“

your study is not about compliance face mask but about performance in face mask, why you mention it?  „According to the results of a survey conducted with 4.8 million people,[8,9] 44% of responses to the question “Why don’t you wear a mask?” were “because it is uncomfortable”. When asked what makes the mask uncomfortable, many factors were reported, such as dizziness, breathlessness, sweating, and nausea.

Answer for what? „A number of previous studies have attempted to evaluate the effect of wearing masks, however, there has been no clear answer.“

Point after 
]13]. „While several studies have considered high intensity exercise, these 52

were focused on parameters such as heart rate and blood pressure[12] or oxygenation index.13] 

Round 2

Reviewer 2 Report

The manuscript has been significantly improved. Several methodological elements were better explained, but I still consider the small sample size to be a significant limitation.
